



# Potential Impact of Landslide and Debris Flow on Climate Extreme - A Case Study of Xindian Watershed in Taiwan

Shih-Chao WEI[1], Hung-Ju SHIH[2], Hsin-Chi LI[2], Ko-Fei LIU[1]

[1] Department of Civil Engineering, National Taiwan University, Taipei 10617, Taiwan
[2] National Science and Technology Center for Disaster Reduction, New Taipei City 23143, Taiwan

*Correspondence to*: Hsin-Chi LI (hsinchi@ncdr.nat.gov.tw)

**Abstract.** Sedimentary produced and transported in mountainous area under extreme rainfall by climate change is a challenged issue in recent years, especially in a watershed scale. The scenario approach with coupled simulation by different models could be one of a solution for further discussion under warming climate. With properly model selection, the simulation of projected

rainfall, landslide, and debris flow are integrated by fully connection between models. Moreover, a case in Xindian watershed upstream the capital of Taiwan is chose for studying, and two extreme scenarios in late 20th and late 21st century are selected for comparison on changing climate. With sequent simulation, the chain process and compounded disaster can be considered in our analysis. The potential effects of landslides and debris flows are compared between current and future, and the likely impact in selected watershed are discussed under climate extreme. Result shows the unstable sediment volume would enlarge

29% in terms of projected extreme event. The river bed may have strong variation by serious debris flow and increase about 10% elevation in main channel. These findings also highlight the increasing risk in stable water supply, isolated village effect, and other secondary disaster in this watershed. A practical reference could be provided by some critical information in our result for long-term adapted strategies.

## 1 Introduction

In recent years, the frequency and magnitude of disaster associated with climate extreme have increased in different evidences. A major concern between climate change and disaster impact becomes important for scientists, policymakers, and the public (IPCC, 2012). Thereby, many researchers has paid their attention to build projected future climate events or explore potential hazard effect of extreme weather.

With the statistics evaluation of World Bank, Taiwan is listed in a high risk disaster area, especially on the slope-land disasters

(Dilley et al., 2005). Therefore, we focus our attention on the slope-land disaster associated with climate change. In addition, a watershed scale is selected to explore the potential effect and to discuss the likely impact of slope-land problem under climate extreme. In climate extreme, previous studies show that the changes in temperature and precipitation will likely affect geo-hazards on slope land such as landslide or debris flow in each local area (Hsu et al., 2011; Chen et al., 2011). Although temperature can play an important role on slope-land problem such as snowmelt or glacier wasting (Rebetez et al., 1997;



Chiarle et al., 2007; Stoffel et al., 2014), this is not included in this study due to Taiwan is located at low latitude area without snow and glacier in wet season. Therefore, the precipitation becomes the most important factor associated with climate change in Taiwan and it will be the only triggering factor of climate change in the following discussion.

In Taiwan, the major series geo-hazards on slope land can be separate into two parts: landslide, and debris flow. To link these hazards with meteorological properties, the scenario approach is one of the popular way used in recent years. Some of the potential effects on landslide has been explored by given scenarios between current and future (Buma and Dehn, 1998; Collison et al., 2000; Cozier, 2010; Shih et al., 2015). Similarly, the frequency of debris flow has been investigated by projected future rainfall (Rebetez et al., 1997; Stoffel et al., 2014). However, the above-mentioned researches are focus on the relation between meteorological properties of climate change and one of the consequent hazard such as rainfall-landslide, or rainfall-debris flow. But on slope land, different kind of single hazards usually have strong linkage with each other, the chain reaction usually resulted in compound hazards from each single hazards (Chen et al., 2011; Liu et al., 2013). For example, landslide mass is also one of the triggering factor of debris flow (Takahashi, 1981; Iverson, 1997). Therefore, a comprehensive assessment becomes a challenge in the chain reaction between each single hazards on slope land.

By using the given rainfall scenario, the consequent landslide, soil erosion, debris flow, sediment transport, or turbidity in reservoir, etc. can be simulated by different theories or numerical programs so far. However, the theories of different physical phenomena such as sediment production or transport process are discrepancy. Therefore, some researchers start to apply suitable numerical techniques in different physical phenomena, and combine them as a chain by linking a model output to the other model input (Chen et al., 2011; Hsu et al., 2012; Liu et al., 2013; Wu et al., 2016; Fan et al., 2017; Li et al., 2017). With integrated simulation, Chen et al. (2011) has discussed the compound disaster including landslide, debris flow, small-scale floods in past real event. Liu et al. (2013) combined landslide, debris flow, and sediment transport in a catchment scale to simulate the turbidity in the reservoir by using an assumed scenario. Likewise, Wu et al. (2016) also assessed a comprehensive disaster impact including landslide, debris flow, flooding, and coastline disasters in whole watershed. Li et al. (2017) extended this kind of simulation results on further potential loss assessment. The models can be altered in different physical processes but should be properly selected for their discussion issue. Significantly, the chain reaction only can be demonstrated in fully connection between models' output and input.

In the present study, the scenario approach will be used in a selected watershed. With given scenarios, the integrated simulation on landslides and debris flows will be adopted for discussion of potential effect or likely impact under changing climate.

## 2 Methodology

The sediment from production to transport can be regarded as a chain reaction. For example in Taiwan, heavy rainfall triggered by typhoon usually leads to high density of landslides on slope land. When the loose landslide deposit mix with runoff or dammed-lake water, debris flows will occur and flow along gully to downstream. It means the chain reaction of sediment cannot be ignored and should be considered in the discussion of sediment in mountainous area, especially in large scale





discussion. In this process, different physical mechanisms could be simulated by combination of different models (Chen et al., 2011; Hsu et al., 2012; Liu et al., 2013; Wu et al., 2016; Fan et al., 2017; Li et al., 2017) such as rainfall, landslide, debris flow, etc. The detail of each parts will be introduced in the following.

## 2.1 Rainfall Scenarios

In recent decades, climate projection for different periods are widely accessible by General Circulation Models (GCM), and it is useful for studying the sequence issues of changing climate at global scale such as frequency of natural hazards, assessment of risk, strategy of adaptation. However, the resolution of GCMs are too course to simulate data for further application in hydrology or agriculture at local scale. For example in typhoon rainfall, the GCM result still cannot describe the small-scale weather patterns in our local area and it also cannot assess in daily or hourly timescale. To link the simulation scale between

atmosphere and hydrology, downscaling techniques (statistical or dynamic downscaling) are suggested accomplishing this demand. In Taiwan, these techniques also has been applied in climate projection simulation by Taiwan Climate Change Projection and Information Platform (TCCIP) funded by the Ministry of Science and Technology, and the downscaling dataset are freely provided in the official website of TCCIP.

For the data provided by TCCIP, the Atmospheric General Circulation Model 3.2 (AGCM 3.2) developed by the

Meteorological Research Institute (MRI), Japan Meteorological Agency (JMA) is used for global climate simulation at 20 km horizontal resolution (Mizuta et al., 2012). Meantime, the observed sea surface temperature are considered as lower boundary condition by Coupled Model Intercomparison Project Phase 5 (CMIP5). With the initial state and boundary conditions from the result of MRI-AGCM 3.2, the dynamical downscaling dataset at 5 km horizontal resolution are simulated by Weather Research and Forecasting Modeling System (WRF) (Skamarock et al., 2008) which is developed by the U.S. National Center

for Atmospheric Research (NCAR). Because the underestimate in total rainfall is still found between observations and WRF results, the quantile mapping method is adopted for bias correction in these datasets (Su et al. 2016).

Because the climate uncertainties are difficult to assess, the worst case of Representative Concentration Pathway 8.5 (RCP 8.5) scenario issued by IPCC Fifth Assessment Report (IPCC, 2013) is selected to analyse for engineering purpose. According to the RCP 8.5 scenario, the projection rainfall data in late 20th century (1979-2003) and late 21st century (2075-2099) are

simulated from TCCIP, and the hourly rainfall at 5 km horizontal resolution are provided for end user. However, the spatial resolution in 5 km is still too large for simulation of landslide or debris flow. To satisfy the precision for further discussion, the spatial interpolation from 5 km to 40 m is made for selecting scenarios, and used as the input of landslide simulation.

## 2.2 Landslide Inventory Simulation

Potential landslide simulation model is used to evaluate the probability of landslide or assess if it will slide or not. According

to different theories, it can be separated into two major branches: statistical approach such as binary regression model (Chang and Chiang, 2009), and physical approach such as slope stability analysis (Baum et al., 2008; Montgomery and Dietrich, 1994). Because the rainfall input in each grid cell are non-homogeneous in space and time. A grid-based model could be practical for



connection between rainfall and landslide. Therefore, a physical approach, the Transient Rainfall Infiltration and Grid-based Regional Slope-Stability Model (TRIGRS), is selected for assessing landslide area.

The TRIGRS is an inventory of shallow landslide simulation program developed by USGS (Baum et al., 2008), and it has been widely used in several case studies of shallow landslide (Salciarini et al., 2006; Godt et al., 2008; Park et al., 2013; Shih et al.,

2015). As TRIGRS is a shallow landslide simulation program, the deep-seated landslide is not available in our discussion. With infinite slope stability hypothesis, the potential slope failure can be determined by the ratio of resisting basal Coulomb friction to gravitationally induced driving stress. The ratio is called Factor of Safety (FS) and can be expressed as

$$FS(Z,t) = \frac{\tan\phi}{\tan\alpha} + \frac{C - \psi(Z,t)\gamma_w \tan\phi}{\gamma_s d_{LZ} \sin\alpha \cos\alpha} \ , \tag{1}$$

where $\phi$ is soil internal friction; $C$ is cohesion; $\gamma_w$ and $\gamma_s$ are unit weight of water and soil; $d_{LZ}$ is the depth of the

impermeable lower boundary or soil thickness; $\alpha$ is slope angle; $\psi(Z,t)$ is the pore water pressure with vertical soil thickness $Z$ at time $t$, and it is calculated by a linearized solution of one-dimensional Richard's equation (Iverson, 2000; Baum et al., 2008) below

$$\psi(Z,t) = [Z - d]\beta$$
$$+2\sum_{n=1}^{N}\frac{I_{nZ}}{K_Z}H(t-t_n)[D_1(t-t_n)]^{\frac{1}{2}}\sum_{m=1}^{\infty}\left\{ierfc\left[\frac{(2m-1)d_{LZ}-(d_{LZ}-Z)}{2[D_1(t-t_n)]^{1/2}}\right]+ierfc\left[\frac{(2m-1)d_{LZ}+(d_{LZ}-Z)}{2[D_1(t-t_n)]^{1/2}}\right]\right\}$$
$$-2\sum_{n=1}^{N}\frac{I_{nZ}}{K_Z}H(t-t_{n+1})[D_1(t-t_{n+1})]^{\frac{1}{2}}\sum_{m=1}^{\infty}\left\{ierfc\left[\frac{(2m-1)d_{LZ}-(d_{LZ}-Z)}{2[D_1(t-t_{n+1})]^{1/2}}\right]+ierfc\left[\frac{(2m-1)d_{LZ}+(d_{LZ}-Z)}{2[D_1(t-t_{n+1})]^{1/2}}\right]\right\}$$
$$\tag{2}$$

where $d$ is the vertical depth of steady-state water table, $\beta = \cos^2\alpha - (I_{ZLT}/K_S)$, where $K_S$ is the saturated hydraulic

conductivity and $I_{ZLT}$ is the initial surface flux, $I_{nZ}$ is the surface flux of a given intensity for the $n^{th}$ time interval (i.e. rainfall input), $D_1 = D_0 \cos^2\alpha$, where $D_0$ is the saturated hydraulic diffusivity, $N$ is the total number of time intervals, and $H(t-t_n)$ is Heaviside step function. The $ierfc$ is the first integral of complementary error function and can be expressed as $ierfc(\eta) = \pi^{-1/2}\exp(-\eta^2) - \eta erfc(\eta)$ .

During the TRIGRS simulation, the FS of each grid is larger than 1, i.e. the infinite slope is stable, in the beginning. With the

rainfall and infiltration, the FS will decrease by the increment of pore water pressure. The unstable grid or failure of an infinite slope is occurred once the FS less than 1, and it will be regarded as potential landslide area. With soil thickness $d_{LZ}$ in each unstable grid, the potential landslide volume could be further evaluated for debris flow simulation input.

### 2.3 Debris Flow Simulation

For debris flow assessment, numerical simulation is the most popular approach in recent years (O'Brien et al., 1993; Hutter et

al., 1995; Hungr et al., 1995; Sassa et al., 2004; Liu and Huang, 2006; Nakatani et al., 2008; Armanini et al., 2009; Christen at., 2010). Although different kind of numerical programs were developed from different theories, the governing equations





comes from mass and momentum conservation. With practical application of input, these models could be divided into two branches such as hydrological-based with a calibrated hydrograph at a specified inflow location (O'Brien et al., 1993; Nakatani et al., 2008; Armanini et al., 2009), and geologic-based with initial mass distributed on its source area (Hutter et al., 1995; Hungr et al., 1995; Sassa et al., 2004; Liu and Huang, 2006; Christen et at., 2010). To link debris flow simulation with landslide

result, a geologic-based model is more useful for application. In this study, we apply Debris-2D (Liu and Huang, 2006) for simulating the debris-flow transport process.

The Debris-2D has been widely applied in real debris-flow cases (Liu et al., 2009, Tsai et al., 2011, Wu et al., 2013). In Debris-2D, the debris flow is treated as a single phase non-Newtonian fluid. A three-dimensional constitutive relation generalized by Julien and Lan (1991) is adopted in this model and the depth-averaged governing equations to the leading order (Liu & Huang,

2006) are

$$\frac{\partial H}{\partial t} + \frac{\partial (uH)}{\partial x} + \frac{\partial (vH)}{\partial y} = 0 , \tag{3}$$

$$\frac{\partial (uH)}{\partial t} + \frac{\partial (u^2 H)}{\partial x} + \frac{\partial (uvH)}{\partial y} = -gH \cos\theta \left( \frac{\partial B}{\partial x} + \frac{\partial H}{\partial x} \right) + gH \sin\theta - \frac{1}{\rho} \frac{\tau_0 u}{\sqrt{u^2 + v^2}} , \tag{4}$$

$$\frac{\partial (vH)}{\partial t} + \frac{\partial (uvH)}{\partial x} + \frac{\partial (v^2 H)}{\partial y} = -gH \cos\theta \left( \frac{\partial B}{\partial y} + \frac{\partial H}{\partial y} \right) - \frac{1}{\rho} \frac{\tau_0 v}{\sqrt{u^2 + v^2}} , \tag{5}$$

where $H$ is flow depth; $B$ is bed topography; $u$ and $v$ are depth-averaged velocities in $x$ - and $y$ -direction respectively; $\theta$

is bed slope; $\tau_0$ and $\rho$ are debris-flow yield stress and density; $g$ is gravitational acceleration. Without consideration of bottom erosion and deposition effect, the yield stress becomes the dominant bottom stress. By applying Eq. (3)~(5), the $H$ , $u$ , and $v$ can be calculated with an initial stationary debris pile $H$ . In addition, all debris flow obeys a starting condition (Liu and Huang, 2006) and only flows once pressure and gravitational effects exceed the yield stress effect.

Based on Eq. (3)~(5), the input data are topography $B$ , initial debris flow depth $H$ , and yield stress $\tau_0$ . The topography is

decided by Digital Elevation Model (DEM) which is the same as landslide simulation. Because the climate extreme is considered, there would be enough rainfall to initialize all dry debris produced from landslide. With the landslide area simulated by TRIGRS and its corresponding soil thickness $d_{LZ}$ in each grid calls, the initial debris flow depth $H$ can therefore be determined by the following relation (Liu and Huang, 2006; Liu et al., 2009)

$$H = d_{LZ}/C_{d\infty} , \tag{6}$$

where the $C_{d\infty}$ is equilibrium concentration (Takahashi, 1981) below

$$C_{d\infty} = \frac{\rho_w \tan\theta}{(\rho_s - \rho_w)(\tan\phi - \tan\theta)} , \tag{7}$$

where $\rho_w$ and $\rho_s$ are density of water and sediment; $\phi$ is internal friction angel; $\theta$ is average creek bottom slope.



## 2.4 Integrated Simulation Process

With above introduction, an integrated simulation process is built as shown in Fig. 1. According to Fig. 1, rainfall events are simulated by MRI-AGCM3.2, downscaled with WRF, and modified with bias correction. To accomplish the research demand on discussion of climate change, two extreme rainfall scenarios from different periods are collected to compare the difference.

With rainfall input and other corresponding parameters, the potential landslide area are simulated by TRIGRS and the landslide inventory maps are made for different scenarios. Because the sediment production during a rainfall event is dominated by landslide mass (Hsu et al., 2012), the soil erosion on slope land is ignored in our discussion. Under extreme rainfall and loose landslide mass, we assume sediment will transport by debris flow from upstream catchment. Accordingly, all landslide mass is considered as the input of debris flow and simulated by Debris-2D. The chain effect may be reacted by this integrated

simulation in a watershed point of view and the compounded disaster between current and future can be compared in terms of climate change.

## 3 Case Study: Xindian Watershed

### 3.1 Study Area

Xindian watershed is located in the upstream of capital, Taipei, northern Taiwan. Xindian river is one of the three major

tributaries into the Tamsui River and it is one of the main source of drinking water for Taipei city and New Taipei city. According to the Taipei City Running Water Center, over 4 million Taipei residents obtain 97% of their drinking water from this river. The main tributary of Xindian river are Nanshi river and Beishi river, as shown in Fig. 2. Comparing this two tributaries, the geological condition in Nanshi River catchment is more fracture than Beishi river catchment, and the historical landslide are concentrated along Nanshi River (Wei et al., 2015, Wu et al., 2016). So we focus our attention on the Nanshi

River catchment and ignore the catchment beyond Feitsui reservoir.

In Fig. 2, the whole study area is 49,000 ha. The villages are major along Nanshi river such as Wulai, Xinxian, and Fushan including 2716, 622, and 739 inhabitants, respectively. In this area, it can be found the high variation of elevation, and can be seen lots of canyon-like topography along the bank of Nanshi river. The study area are mainly located in Tatungshan Formation (Tt), Szeleng Sandstone (Em), Kangkou Formation (Kk), Mushan Formation (Ms), Tsuku Formation (Tu), and its content are

major from sand stone, argillite, slate, shale, siltstone. The age of geological setting are between Holocene and Eocene. Lots of folds and faults are distributed in this area.

Because soft and fractured geological condition in Nanshi river catchment, the geo-disaster and its sediment problem becomes an important issue in this area (Wei et al., 2015; Wu et al., 2016). For example, the Typhoon Soudelor during 7th Aug. and 9th Aug. in 2015 has brought heavy rainfall in this area. The maxima accumulated precipitation reached 792 mm at Fushan

meteorological station. Besides, the maxima accumulated rainfall in 3 hrs, 6 hrs, and 12 hrs at Fushan meteorological station were beyond 200-years return period (Wei et al., 2015), and came to 253 mm, 442 mm, and 655 mm, respectively. The maxima




intensity in this event got up to 95 mm/hr and the average intensity on 8th Aug., 2015 also reached 80 mm/hr. Lots of landslide and debris flow were triggered by this short period and high intense rainfall. The total landslide volume was estimated as $9.8 \times 10^6$ m³ and the sediment delivery rate was evaluated about 40.98% in Nanshi river. (Wu et al., 2016). The regional landslide disasters induced the closure of roads and the debris flow from tributary also increased the concentration in the

Nanshi river. According to Taipei Water Department, the peak turbidity came to 39,300 NTU at 08:00, 8th Aug., 2015 and it took 42 hours to reduce to 3,000 NTU (the limit of turbidity of water treatment plant). Therefore, the stable water supply was broken and the quality of water was also influenced during 9th Aug. and 12th Aug, 2015 (4 days) in Taipei area.

As mentioned above, the influence of sediment related hazard not only occurs in Xindian watershed but affects in downstream Taipei city. For long-term city planning aspect, it is urgent to realize the whole situation and establish adapted strategies under

climate change. Accordingly, we will apply the above-mentioned integrated simulation to assess the potential impact in this watershed.

### 3.2 Extreme Rainfall Scenarios

The projection rainfall in late 20th century (1979-2003) and late 21st century (2075-2099) collected from TCCIP are chose for comparison in our study area. Because the bias correction with observation is conducted in previous research (Su et al., 2016),

we will directly use these data without calibration. With engineering purpose, the worst cases (i.e. the rank 1 rainfall event) in the late 20th and 21st century are selected for comparison and we will use scenario 1 and scenario 2 for abbreviation. The accumulated rainfall distribution are shown in Fig. 3a. With representative data at Fushan meteorological station, the maxima accumulated rainfall are 911.4 mm in 61 hr and 1531.1 mm in 40 hr, and the maxima intensity comes to 49.7 mm/hr and 125.8 mm/hr, respectively. For both scenarios, the temporal and spatial resolution are in 1 hr and 5 km, respectively. The spatial

interpolation from 5 km to 40 m is adopted for landslides and debris flows simulation purpose.

### 3.3 Landslide Inventory Simulation

Based on Eq. (1) and Eq. (2), the TRIGRS input data of each grid cell is separated into four parts: rainfall intensity, topographic information, soil parameters, and hydraulic parameters. The rainfall intensity $I_{nZ}$ (mm/hr) presented in previous section are directly used. The topographic information, slope $\alpha$ and flow aspect, are derived from DEM at 40 m resolution. The soil

parameter and hydraulic parameters such as $C$, $\phi$, $\gamma_s$, $K_S$, and $D_0$ could be calibrated by past events or cited from past investigation (Central Geological Survey, 2010). The soil thickness $d_{LZ}$ is calculated by empirical slope-depth relation (Khazai and Sitar, 2000; National Science and Technology Center for Disaster Reduction, 2011). Without considering antecedent precipitation, we assume the initial depth of steady-state water table $d$ is the same as soil thickness $d_{LZ}$ and the initial infiltration rate for soil is $10^{-8}$ (m/s) (Chen et al., 2005).

Because the parameters, $C$, $\phi$, $\gamma_s$, $K_S$, and $D_0$ are subject to geology, the Modified Success Rate (*MSR*) (Huang and Kao, 2006; Shih et al., 2015) is introduced for calibration and validation with different geologic zones, as shown below

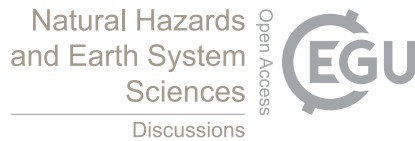



$$MSR(\%) = \frac{1}{2}\frac{N_1}{N_1 + N_2} + \frac{1}{2}\frac{N_4}{N_3 + N_4}. \qquad (8)$$

$N_1$ and $N_2$ denotes the area of $FS < 1$ and $FS > 1$ for those historical landslide area, respectively. Likewise, $N_3$ and $N_4$ presents the area of $FS < 1$ and $FS > 1$ for those historical non-landslide area. The success rate of landslide and non-landslide can be both specified in Eq. (8). The unit of $N_1 \sim N_4$ is calculated by slope-unit. A success prediction is defined while $MSR >$

70%. With the objective function, $MSR$, the parameters in each zones could be optimized by the rainfall events provided by Central Weather Bureau and historical landslide data provided by Central Geological Survey. However, the historical landslide data are only updated by year, the union of representative rainfall events in each year are used for calibration. The calibrated $MSR$ results from 2008 to 2012 are between 84% and 88%, and the calibrated parameters are shown in Table 1. By these parameters, the landslide of typhoon Soudelor in 2015 was validated in a good agreement with $MSR = 91\%$.

During simulation, the potential landslide areas are increasing as accumulated rainfall with a time delayed. The growing rate of accumulated landslide ratio in scenario 2 is faster than in scenario 1 due to the effect of rainfall intensity. The stable time with maxima accumulated landslide ratio for scenario 1 and scenario 2 are within 65 hrs and 40 hrs, respectively. With potential landslide area (FS<1) and its soil thickness $d_{LZ}$ in each grids, the landslide inventory maps are shown as Fig. 3b, and used for debris flow simulation.

**3.4 Debris Flow Simulation**

In Debris-2D simulation, the input data could be separated into three parts: topography (DEM), initial debris-flow mass, and yield stress. For initial debris flow mass, it is calculated by landslide inventory maps with Eq. (6) and (7). Because debris flows are triggered by an extreme rainfall and loose landslide mass, all landslide mass is assumed to be transformed as initial debris-flow mass. Moreover, the high concentration is considered to be occurred, and the maxima $C_{d\infty} \leq 0.603$ (Liu and Huang 2006;

Liu et al. 2009) is used for practical estimation in Eq. (6) and (7). However, we cannot predict when the debris flow will occur after slope failure, so the beginning of debris flows are assumed in the same starting time. For yield stress, it is calibrated by plan test (Liu & Huang 2006) with several samples and the value of 800 Pa is used for simulation.

The final flow depth of debris flows in both scenarios are shown in Fig. 3c. For both scenarios, the upstream Hapen river debris flows cannot transport to downstream due to the meandering creek. The landslide along the downstream Hapen river and

Daluolan river are major deposited before the junction of Zhakong river, Daluolan river, and Happen river. The contribution of flow depth from Daluolan river and Happen river are insusceptible for the debris flow of Zhakong river or Nanshi river. The front of Zhakong river debris flow deposits before a sharp turn in the upstream of Nanshi river in scenario 1, but it reaches the tail of Nanshi river debris flow in scenario 2. The Nanshi river debris flows are deposited before the junction of Nanshi river and Tonghou river.




## 4 Result and Discussion

### 4.1 Potential Effect on Natural Hazards

In this integrated simulation process, the top 1 rainfall events of the worst RCP 8.5 climate scenario are selected for initial input. So the discussion leads to the influence of climate extreme and the comparison of different extreme scenarios. With these rainfall scenarios, the maxima error of landslide area simulated by TRIGRS could be estimated as 16% because the MSR are between 84% and 91% (see Table 1). Moreover, the empirical landslide depth is the input of TRIGRS and it is included in the MSR validation. Therefore, we can simply use the error of landslide area to estimate the error of landslide volume, even though the surveyed data of landslide volume is not available. For debris flow, the source landslide volume is the most sensitive parameter in Debris-2D simulation, and the volume uncertainty in 20% could be resulted in 2.76% variation on its deposition front (Tsai et al., 2011). So the uncertainty of deposition front could be roughly estimated as 2.2% in selected rainfall scenarios. In these two scenarios, the grid-averaged maxima accumulated rainfall are 852.56 mm in 61 hr and 1255.14 mm in 49 hr, and the grid-averaged maxima accumulated rainfall in 24 hr comes to 484.84 mm and 1085.05 mm, respectively. Comparing these two scenarios, we found the grid-averaged accumulated rainfall increase 402.58 mm but the duration decrease 12 hr. Based on scenario 1, the increment of grid-averaged accumulated rainfall and decrement of duration between these two scenarios are 47.22% and 19.67%, respectively.

By using these rainfall scenarios, the landslide area in both scenarios are simulated as $1.67 \times 10^7$ m² and $2.14 \times 10^7$ m² (3.42% and 4.37% of the whole watershed), respectively. With empirical slope-depth relation, the landslide volume are estimated as $7.12 \times 10^7$ m³ and $9.19 \times 10^7$ m³. The total debris flow volume transformed by Eq. (11) are $1.18 \times 10^8$ m³ and $1.52 \times 10^8$ m³. Based on scenario 1, the increment of landslide area, landslide volume, and debris flow volume between these two scenarios are 28.14%, 29.07%, and 28.81%, respectively.

In debris flow simulation, the major accumulation are deposited along Nanshi river and Zhakong river, and the longitudinal profile of accumulation for both scenarios are shown in Fig. 4. In Fig. 4, the part within and beyond 15 km belongs to Nanshi river and Zhakong river, and its corresponding average bed slope are 16.85(m/km) and 31.96(m/km), respectively. Because the mild slope alone river bed, the landslide along river bank are directly accumulated on its foot area, and the debris flow from tributary are decelerated and deposited as well. The abundant sediment results in strong variation of river geomorphology in both scenarios. By comparing these two scenarios, the average river bed elevation increase 12.32% on Nanshi river (0~15 km) and 7.74 % on Zhakong river (15~25 km).

### 4.2 Likely Impact on Compounded Disaster

In this study, the worst cases of projected events are selected for discussion of climate extreme and environmental variation. Therefore, the simulation results are not enough and not suitable for doing local risk assessment in each village or tribe. However, the extreme scenarios may provide some critical information and reliable impact in terms of a whole watershed view. So the compounded disaster and its sequent impact are discussed in a regional level here.





The catastrophic slope-land hazard are occurred under extreme rainfall events no matter which scenario we considered. The large amount sedimentary material is supposed to be produced by landslide and to be transported by debris flow or mud flow with sufficient water. The average accumulated rainfall of scenario 1 and scenario 2 are 1.55 and 2.28 times as heavy as Typhoon Soudelor, and the estimated landslide volume in scenario 1 and scenario 2 are 7.26 and 9.37 times as large as Typhoon Soudelor. From the experience of Typhoon Soudelor in 2015, the high turbidity is believed to be occurred in downstream of Nanshi river during our scenarios and the stable water supply will be broken at last three weeks if no measures are taken for prevention.

Between 2 km and 4 km in Fig. 4, the landslide along the right bank of Nanshi river are directly block the river and gradually become a dam within 4km. The front of debris flow are stop at 0 km in both scenarios, as shown in reach a of Fig. 4. The downstream Wulai Village would be exposed in a high risk of flooding and dam breaking after debris flow. For Xinxian Village in reach b of Fig. 4, all village are not covered by debris flow. Although there are two potential debris flow torrents at 4.5 km and 5 km (downstream and upstream of Xinxian Village) listed by Soil and Water Conservation Bureau in Taiwan, the debris flows here are insusceptible due to past effort of countermeasure. Between 6 km and 9 km in reach c of Fig. 4, the landslide along the left bank of Nanshi river causes the road closure in both scenarios. Because it is the only way to reach Fushan Village, the isolated effect would be forced to happen once the road closure occurs. In reach d of Fig. 4, the Fushan Village are also covered by landslide. Therefore, the residents in Fushan Village should be forcibly retreated before the events. As the accumulation in Fig. 4, a sequence dammed lake may occur along Nanshi river and Zhakong river after the extreme event like our scenarios. The secondary disasters by alluvium along river should be continuously concerned after event such as high turbid river water, stream-triggered debris flow, dam breaking torrent, etc.

**5 Conclusion**

With extreme climate projected scenarios, we proposed an integrated physical simulation process to analyse the potential effect or likely impact of landslides and debris flows in a watershed point of view. The Xiandian watershed upstream capital of Taiwan is selected for discussion.

Thanks to TCCIP platform, two extreme rainfall scenarios in late 20th century (1979-2003) and late 21st century (2075-2099) simulated by MRI-AGCM 3.2 and downscaled with WRF are collected for landslide simulation. The potential landslide area are simulated by the TRIGRS model and the parameter in TRIGRS are calibrated and validated by past events. With potential landslide area, the landslide volume is estimated by empirical slope-depth relation. Than the catastrophic debris flows are considered to occur under the extreme rainfall and large amount of landslide volumes, and simulated by Debris-2D model. In this simulation process, the uncertainty of landslide volume and debris flow deposition front are roughly estimated as 16% and 2.2%.

There is no surprise that landslide volume, debris flow volume, and river bed elevation are increased in an increment of accumulated rainfall triging by climate change. For comparison of these two scenarios, the grid-averaged accumulated rainfall



increase 47.22% but the duration decrease 19.67%. With increasing precipitation, the estimated landslide volumes and debris-flow volumes are in an increment of 29.07% and 28.81%, respectively. The landslide along river bank and the tributary debris flow also cause a serious increment of river bed. The average raised ratio of river bed along Nanshi and Zhakong river are 12.32% and 7.74 %, respectively.

In Xindian Watershed, the possible disasters leads to the influence of water supply, isolated effect of tribe, and increasing risk of secondary disasters. With our practical assessment, the loss of each disasters and its corresponding countermeasures could be further examined or quantified (Liu et al. 2009). The policy-making and the long term strategy in different aspect might be made thereafter.

## Acknowledgments

This research was supported by the Ministry of Science and Technology of Taiwan under MOST 105-2625-M-865-001. The CERG-C program of the University of Geneva is thanked for providing the expertise to carry out this project.

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


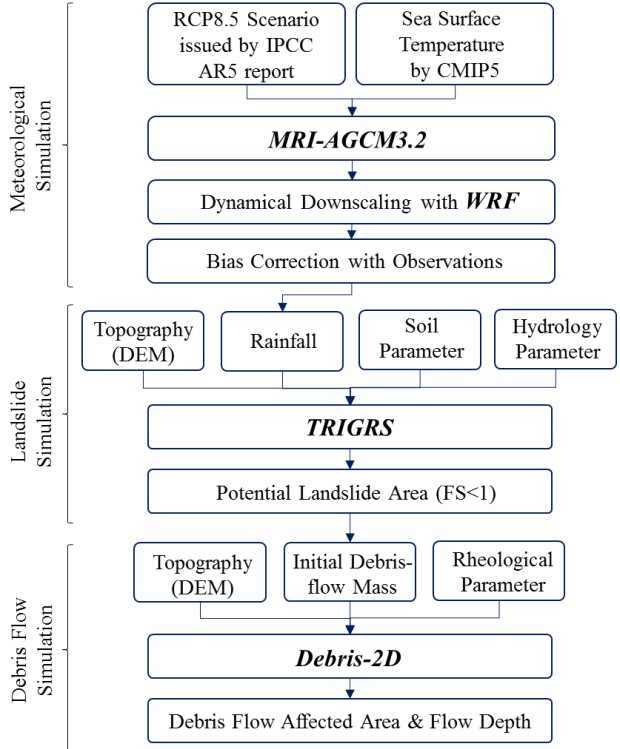

Figure 1: Integrated Simulation Process.







**Figure 2: Topography (shaded relief) and location of Study Area**









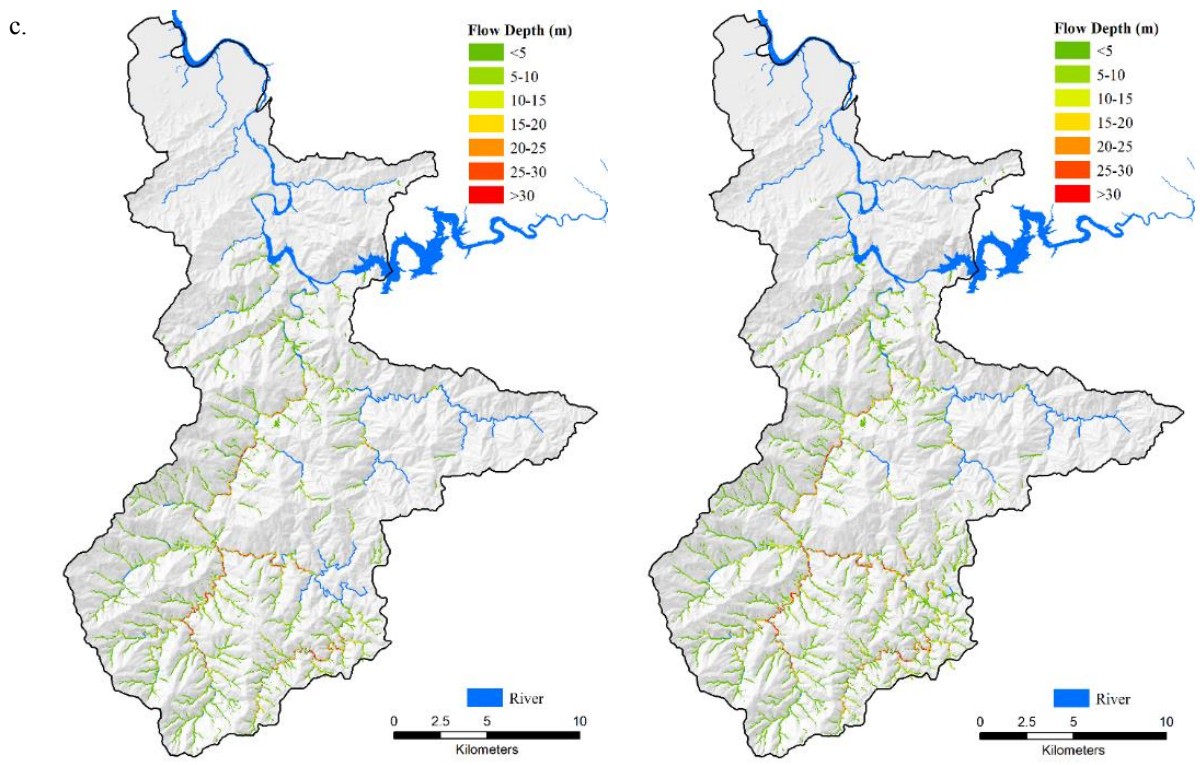

**Figure 3: Simulation Results. a. Accumulated rainfall distribution; b. Potential landslide area; c. The final flow depth of debris flow; the left and right figures are scenario 1 and scenario 2, respectively.**

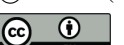



**Figure 4: Longitudinal profiles of debris-flow accumulation along river and its corresponding aerial photos in different regions; the reach of profile in top right figure are along the red dots with 1 km interval in top left figure. The black windows in top left figure are region of interest. The simulation results in each windows are highlighted by gray background in top right figure and shown in below figure with aerial photos provided by National Land Surveying and Mapping Center (NLSC), Taiwan.**



**Table 1 The parameters used in TRIGRS**

| Geologic Time | Name (abbr.) | $C$ (kPa) | $\phi$ (°) | $\gamma_s$ (kN/m³) | $K_S$ ($10^{-6}$ m/s) | $D_0$ ($10^{-6}$ m²/s) | Description (Ref: Central Geological Survey in Taiwan) |
|---|---|---|---|---|---|---|---|
| Holocene | Alluvium (a) | 10.5 | 34 | 19.5 | 29 | 8800 | Gravel, sand, and mud |
| | Terrace Deposits (t) | 6.5 | 30 | 23 | 0.7 | 220 | Gravel, sand and clay |
| Pleistocene | Lateritic Terrace Deposits (lt) | 35 | 30 | 18.6 | 0.8 | 800 | Red earth, lateritic gravel, sand, intercalated with sand and silt lentils |
| Miocene | Mushan Formation (Ms) | 16.8-28.8 | 32.0-36.0 | 27.5 | 10 | 2000 | Alternations of sandstone and shale, intercalated with coal seams |
| | Nanchuang Formation (Nc) | 23.5 | 34.5 | 27.5 | 10 | 2000 | Alternations of sandstone and shale, intercalated with coal seams |
| | Nankang Formation (Nk) | 29 | 36 | 27.5 | 10 | 2000 | Sandstone, siltstone, and shale |
| | Piling Shale (Pi) | 19.9-27.4 | 32.0-35.0 | 24.8 | 10 | 2000 | Shale with intercalated sandstone |
| | Shihti Formation (St) | 24.1-30.1 | 32.0-34.0 | 27.5 | 10 | 2000 | Alternations of sandstone and shale, intercalated with coal seams |
| | Tapu Formation (Tp) | 20.9 | 34 | 27.5 | 10 | 2000 | Alternations of muddy sandstone, white sandstone and shale |
| | Taliao Formation (Tl) | 16.3-27.3 | 32.0-36.0 | 27.5 | 10 | 2000 | Shale and sandstone |
| Oligocene-Miocene | Wenshui Formation (Ws) | 16.4-28.9 | 32.0-36.0 | 24.8 | 10 | 2000 | Sandstone and shale interbeds |
| Oligocene | Kangkou Formation (Kk) | 20.6-33.1 | 26.0-31.5 | 25.3 | 20 | 4000 | Argillite or slate intercalated with thin to thick-bedded siltstone |
| | Shuichangliu Formation (Om) | 21.0-33.5 | 29.0-33.0 | 27.5 | 10 | 2000 | Argillite, slate |
| | Tatungshan Formation (Tt) | 19.1-33.0 | 28.0-34.0 | 27.5 | 10 | 2000 | Argillite intercalated with thin to thick-bedded siltstone and fine-grained sandstone |
| | Tsuku Formation (Tu) | 18.0-30.0 | 27.0-30.0 | 25.3 | 10 | 1000 | Alternations of siltstone and argillite |
| Eocene | Chungling Formation (Cl) | 24.8-32.8 | 29 | 25 | 20 | 4000 | Argillite or slate, with thin bedded metasandstone |
| | Hsitsun Formation (Ht) | 22.2-32.6 | 30.5-33.5 | 25 | 10 | 2000 | Thin alternations of argillite and metasandstone |
| | Szeleng Sandstone (Em) | 18.1-32.0 | 28.0-32.0 | 23.5 | 10 | 2000 | Thick-bedded party pebbly quartzitic sandstone, arkosic sandstone and thin alternations, with argillite and thin coal seams on the upper part |

Remarks: The $C$, $\phi$, $\gamma_s$ are calibrated from 2008 to 2012 with $MSR$ = 88%, 87%, 84%, 84%, and 86%, and verified by Typhoon Soudelor in 2015 with $MSR$ = 91%. The representative rainfall events used for calibration are Typhoon Kalmaegi, Typhoon Jangmi, and Typhoon Sinlaku in 2008; Typhoon Parma and Typhoon Morakot in 2009; Typhoon Megi and Typhoon Fanapi in 2010; Typhoon Nanmadol and 1001 Rainfall in 2011; Typhoon Saola in 2012. The $K_S$, and $D_0$ are cited from past investigation (Central Geological Survey 2010)