# Peer review of "Potential Impact of Landslide and Debris Flow on Climate Extreme -A Case Study of Xindian Watershed in Taiwan"

_Natural Hazards and Earth System Sciences, 2017_

## Referee Comment (RC1) · Anonymous Referee #1 · 9 Oct 2017

The authors provide a manuscript dealing with landslides and debris flows in Taiwan, and their possible (coupled) dynamics due to climate change, assessed by a scenario approach.

However, the presentation of the content is very weak in terms of language, which makes it nearly impossible to follow the arguments in a logical way. The authors somehow state that they are assessing hazard chains or compound hazards (landslides triggered by heavy rainfall leading to debris flows), but at the present state of their manuscript I am unable to judge the content from a scientific point of view. Moreover, it remains unclear what exactly is meant by "slope-land processes", maybe hillslope

processes? It is common sense that changes in the process trigger (here: rainfall) will result in changes of process magnitudes and frequencies... assessing the impact would be interesting, also with respect to spatio-temporal dynamics (which is stated in the Abstract but I cannot find it in the Results and Conclusions).

Because of this weakness in language and presentation I simply recommend a rejection of the manuscript – the authors are not encouraged to re-submit this piece of work unless substantial improvements have been made (only then it will be possible to properly review the methods, results and discussion).

---

## Referee Comment (RC2) · Anonymous Referee #2 · 27 Nov 2017

The authors present a research based on scenario simulation for analyzing the impact of shallow landslide and debris flow in a watershed in Taiwan under extreme rainfall. In this research, extreme rainfall is regarded as the main factor for triggering landslides and debris flows. Using rainfall scenarios from climate simulations, the landslide model, TRIGRS, was used for landslide mapping, and the debris flow model, Debris-2D, was then used for debris flow routing. Basically, the content is written in poor English without smooth logic and clear scientific viewpoint. After evaluation, I simply recommend rejection of this manuscript. Some reasons and suggestions are listed in the following.

The simulation concept adopted is not novel, in fact, many similar research on the same

topic have been reported in recent decade. Also, it has been agreed heavy rainfall can trigger abundant landslides and debris flows. However, this manuscript only addresses the production volume and area in different scenarios like others. It is lacking of any new finding or unique merit different from the literature. This means the scientific value of the present manuscript is very low for publication.

The second deficiency is that it exists a lot of weak or unclear explanations throughout the manuscript. Such as, even extreme rainfall is the main factor, its definition or pattern is not explained in detail. Instead, only the rainfall accumulations are drawn without indicating any duration information, and so forth. This similar manner also appears in the parts of landslide and debris flow simulations. Without detailed definition this manuscript has no reference value from the scientific viewpoint or even from the engineering one. Additionally, the manuscript is not technically sound as having many ambiguous terminologies.

The other deficiency is that the authors tried to analyze long profile change influencing by debris flows without considering river flow motion. However, none of scientific evidence is provided for supporting that neglecting river flow is valid in the target watershed. The long profile analysis in 4.1 has no solid scientific basis, and the sequential discussion for compound disasters in 4.2 is meaningless.

According to the evaluation this manuscript is not acceptable for publication at this stage. More clarification and language revision are strongly demanded before future submission.